# Clinical and Epidemiological Characteristics of *Staphylococcus caprae* Infections in Catalonia, Spain

**DOI:** 10.3390/microorganisms13010053

**Published:** 2025-01-01

**Authors:** Javier Díez de los Ríos, María Navarro, Judit Serra-Pladevall, Sònia Molinos, Emma Puigoriol, Noemi Párraga-Niño, Glòria Pedemonte-Parramón, Luisa Pedro-Botet, Óscar Mascaró, Esteban Reynaga

**Affiliations:** 1Internal Medicine Department, Multidisciplinary Inflammation Research Group, Hospital Universitari de Vic, 08500 Vic, Spain; omascaro@chv.cat; 2Fundació Lluita Contra les Infeccions, Infectious Diseases Department, Hospital Universitari Germans Trias i Pujol, Carretera Canyet, 08916 Badalona, Spain; mlpbotet.germanstrias@gencat.cat; 3Escola de Doctorat, Universitat de Vic–Universitat Central de Catalunya (UVIC–UCC), 08500 Vic, Spain; 4Faculty of Medicine, Universitat de Vic–Universitat Central de Catalunya (UVIC–UCC), 08500 Vic, Spain; 5Fundació Institut de Recerca i Innovació en Ciències de la Vida i de la Salut de la Catalunya Central, 08500 Vic, Spain; jserrap@chv.cat; 6Microbiology Department, Multidisciplinary Inflammation Research Group, Hospital Universitari de Vic, 08500 Vic, Spain; mnavarro@chv.cat; 7Faculty of Health Sciences, Universitat de Vic–Universitat Central de Catalunya (UVIC–UCC), 08500 Vic, Spain; 8Microbiology Department, Hospital Germans Trias i Pujol, 08916 Badalona, Spain; smolinos.germanstrias@gencat.cat; 9Epidemiology Department, Multidisciplinary Inflammation Research Group, Hospital Universitari de Vic, 08500 Vic, Spain; epuigoriol@chv.cat; 10Infectious Diseases Unit, Fundació Institut d’Investigació Germans Trias i Pujol, 08916 Badalona, Spain; nparraga@igtp.cat; 11Orthopaedic Surgery and Traumatology Department, Hospital Germans Trias i Pujol, 08916 Badalona, Spain; gpedemonte5@gmail.com; 12Infectious Diseases Unit, Health Sciences Research Institute of the Germans Trias i Pujol Foundation, 08916 Badalona, Spain

**Keywords:** *Staphylococcus caprae*, emerging zoonoses, livestock, goats, Catalonia

## Abstract

*Staphylococcus caprae* is a coagulase-negative staphylococcus commonly associated with animals which can also be a zoonotic human pathogen. To date, there is little data available on *S. caprae* infections. The aim of this study was to characterize the *S. caprae* infections identified in two hospitals located, respectively, in rural and urban areas of Catalonia, Spain. In this retrospective, observational study, data were compiled from clinical records of all *S. caprae* infections diagnosed between January 2010 and December 2023. Over the study period, altogether 31 cases of *S. caprae* infection were identified, with most (23) of these cases occurring in the second half of the period (2017–2023). The mean age of patients was 58.87 ± 20.65 years, and 58.1% were males. Eight patients had had livestock exposure. The most frequent manifestation of infection was skin and soft subcutaneous tissue infections (10; 32.3%), osteomyelitis (6; 19.4%), and joint prosthetic infections (5; 16.1%). All the strains were susceptible to oxacillin, fluoroquinolones, rifampicin, and trimethoprim–sulfamethoxazole. Twenty-two (71%) of the patients required surgical treatment. Only one patient (3.2%) died, because of aortic prosthetic valve infective endocarditis. Skin and soft tissue infections were the most frequently identified manifestations of *S. caprae* infection. Over 75% of the cases occurred in the last six years, and 25.8% involved significant exposure to livestock. Ongoing surveillance is necessary to better understand the prevalence and transmission dynamics of this emerging zoonotic pathogen.

## 1. Introduction

*Staphylococcus caprae* is an animal-associated coagulase-negative staphylococcus [1] that characteristically colonizes the skin and mammary glands of goats [2]. It was first described by Devriese et al. in 1983 [3], based on a strain isolated from goat’s milk [4], and has also recently been isolated from sheep’s milk samples [5].

In humans, *S. caprae* may present as a harmless commensal in the nose, skin, and nails [2]. However, it can also become zoonotic and has been isolated from patients who have been in close contact with goats or sheep, such as farm workers or sheep breeders, and individuals bitten by goats [1]. Other risk factors for acquiring the infection are immunosuppression, obesity, open bone fractures, or tissue trauma.

The mechanisms responsible for the development of *S. caprae* infection remain largely unknown. Watanabe et al. (2018) performed complete genome sequences of three methicillin-resistant *S. caprae* isolates from humans, revealing that *S. caprae* is closely related to *S. epidermidis* and *S. capitis* at the species level, especially in its ability to form biofilms, which may lead to increased virulence during the development of *S. caprae* infections [1].

Cases of *S. caprae* infections have been mainly reported in the following countries: France (n = 48), Spain (n = 16), Canada (n = 12), and USA (n = 11) followed to a lesser extent by Japan, Korea, and other European countries (see Appendix A, which includes cases of *S. caprae* infections reported between 1995–2023).

This microorganism has a special predilection for producing osteoarticular infections [6], and further involvement in cases of bacteremia [7], acute otitis externa [8], urinary tract infection, peritonitis [2], and infective endocarditis [9].

In the autonomous community of Catalonia, the goat and sheep industries are significant; nevertheless, to date, little data are available on *S. caprae* infections in Catalonia. The aim of the present study is to begin to fill this gap by characterizing the 31 cases of *S. caprae* infections that were diagnosed over a thirteen-year period at two hospitals in Catalonia, one located in a rural setting and the other in an urban one.

## 2. Material and Methods

### 2.1. Description of the Study

A retrospective, observational study of *S. caprae* infections was carried out covering the period from January 2010 to December 2023 by examining all the records at the Hospital Germans Trias i Pujol (HGTiP) in Badalona, a city that forms part of the greater Barcelona metropolitan area, and the Hospital Universitari de Vic (HUV), located in the town of Vic, the administrative centre of a rural county. The hospitals serve 800,000 and 156,599 citizens, respectively (Figure 1).

*S. caprae* infection was defined by the isolation of this microorganism in a sample taken from a normally sterile site (blood, urine, synovial fluid, and bone), purulent discharge and orthopedic devices or catheter in a patient with signs and symptoms of infection.

### 2.2. Bacterial Identification and Antibiotic Susceptibility Testing

The microorganism was grown on blood agar plates incubated at 37 °C in a 5% CO_2_ atmosphere for 24–48 h, and bacterial identification was performed using the MALDI-TOF MS Bruker Biotyper (Billerica, MA, USA). The minimal inhibitory concentrations (MICs; μg/mL) of each sample were determined using VITEK^®^2 (Biomerieux, Marcy-l’Étoile, France), the results were then interpreted in terms of antibiotic susceptibility in accordance with EUCAST clinical breakpoints. These diagnostic techniques have been applied in both hospitals since 2010.

### 2.3. Statistical Analysis

The descriptive statistics are as follows: Qualitative variables will be presented as absolute frequency and percentage and quantitative variables as mean, standard deviation, median, minimum, and maximum. The inferential statistics are as follows: χ^2^ (or Fisher’s exact test) was used to analyze the relationship between two variables and Student’s *t*-test (or a non-parametric Mann–Whitney U test) was used to analyze the relationship between a qualitative and a quantitative variable.

## 3. Results

A total of 31 cases of *S. caprae* infection were detected, 16 of them at the HUV and 15 at the HGTiP. Eight cases (25.8%) were detected in the 2010–2016 period and twenty-three (74.2%) in the 2017–2023 period.

The mean of age of the 31 patients was 58.87 ± 20.65 years and a majority were male (18; 58.1%). The mean Charlson Comorbidity Index score was 3.5 ± 3.4. Only one patient (3.2%) was under corticosteroid or other immunosuppressive treatment at the time of infection.

Eight patients had had direct or indirect exposure to livestock, of whom seven (43.8%) lived in the countryside (three had contact with goats, two worked in a slaughterhouse, one patient’s father worked in a slaughterhouse, and another patient’s brother had direct contact with goats), while one patient from an urban area had previously worked as a butcher.

The most frequent manifestation of infection was skin and soft subcutaneous tissue infection (10; 32.3%), osteomyelitis (6; 19.4% → 3 involving toes, 2 the calcaneus, and 1 the tibia), joint prosthesis infection (5; 16.1% → 1 involving hip prosthesis, 1 involving knee prosthesis, and 3 involving osteosynthesis material), bursitis (3; 9.7% → 2 involving elbow and 1 knee), prosthetic valve infective endocarditis (2; 6.5%), urinary tract infection (2; 6.5%), diabetic foot infection (1; 3.2%), tunnelled catheter for haemodialysis infection (1; 3.2%) and otitis media (1; 3.2%).

Most patients required surgical treatment (71%). Only one patient (3.2%) died, due to aortic prosthetic valve infective endocarditis.

Table 1 shows the clinical and epidemiological characteristics of the 31 cases of *S. caprae* infection.

Table 2 shows the summary of individual characteristics of the 31 patients diagnosed with *S. caprae* infection as well as the treatment prescribed.

All of the *S. caprae* strains isolated were susceptible to oxacillin, fluoroquinolones, rifampicin, and trimethoprim–sulfamethoxazole, and 83.9% were susceptible to clindamycin, while only 16.1% were susceptible to penicillin.

Thirteen patients were treated with fluoroquinolones (two of them associated with rifampicin), six patients with amoxicillin/clavulanic acid, two patients with cloxacillin plus rifampicin, and trimethoprim–sulfamethoxazole, clindamycin and linezolid were used for the remaining three patients, respectively. For seven patients, no information was available regarding the antibiotic treatment prescribed.

Figure 2 shows the difference in antibiotic resistance between rural and urban areas.

## 4. Discussion

Although there are almost 70,000 heads of goats registered in Catalonia, goat-farming is more prevalent in several other of the autonomous communities that make up Spain, such as Andalusia and Extremadura [10]. Nonetheless, very little research about *S. caprae* infections in Spain has been published [8,11,12]. Hence, the present study offers a description of a significant number of cases identified in two hospitals in Catalonia. The distribution of cases between the HUV and the HGTiP was relatively even, with a noticeable increase in the number of infections reported at both sites in the latter half of the study period (2017–2023) compared to the earlier half (2010–2016). This could be influenced by the increase in the number of surgeries in recent years as well as the awareness of surgeons to send representative samples for a correct microbiological diagnosis, suggesting that *S. caprae* is likely to become a more prevalent casual infectious agent in the near future.

*S. caprae* infections have been associated with direct or indirect contact with goats or sheep, and indeed in our study we found that seven patients (43.8%) from rural areas had had occupational or environmental exposure and one patient from an urban background had worked as a butcher. This is a higher percentage of cases than that reported in Seng et al. (2014), where 20% of patients in a hospital in south-eastern France were related to close contact with goats or sheep [2], with information about occupational exposure being absent in other reports [11,12,13].

The finding that, overall, 25.8% of patients had had direct or indirect exposure to livestock underscores the zoonotic potential of *S. caprae*. Such findings highlight the importance of considering occupational and environmental factors in the epidemiology of *S. caprae* infections, particularly in rural areas where human–animal interactions are more common.

Interestingly, we found that animal contact was predominantly reported among patients from the HUV, which serves a more rural catchment area, supporting the notion of distinct transmission dynamics between rural and urban settings. In urban areas, alternative transmission routes, such as through contaminated food products (i.e., unpasteurised dairy products or contaminated meat) or the environment (mainly in animal work areas, veterinary hospitals, animal clinics, or animal handling facilities, where the bacteria can survive on contaminated surfaces, such as cages, animal handling equipment or clothing, favouring transmission when people come into contact with these contaminated surfaces or utensils), should be considered. This warrants further investigation to elucidate the mechanisms of infection acquisition, particularly in urban populations.

Previous studies have mainly reported infections involving bone and joints such as prosthetic joint infection [2], osteomyelitis [11], septic arthritis [12], or spondylodiscitis [13]. Similarly, in our study more than 70% percent of cases involved skin and soft subcutaneous tissues, bones, bursa, diabetic feet, and prosthetic joints. Interestingly, we had two cases of infective endocarditis, one involving an aortic bioprosthesis and a mitral native valve and the other a pulmonary bioprosthesis, with only three similar cases reported previously. This distribution reflects the organism’s known propensity for causing biofilm-associated infections, particularly in patients with indwelling medical devices [9]. Seng et al. (2014) similarly identified prosthetic joint infections as a common presentation [2], emphasizing the challenges of managing biofilm-forming bacteria in clinical settings.

The antibiotic susceptibility patterns observed in this study, with high susceptibility to oxacillin, fluoroquinolones, rifampicin, and trimethoprim–sulfamethoxazole, are consistent with previously reported data [14]. In contrast to *S. epidermidis* and *S. haemolyticus* with high percentages of methicillin-resistant isolates of both species [15], other clinically important coagulase-negative staphylococci are mostly less resistant to oxacillin, such as *S. caprae*, according previous studies [14,16,17,18,19]. The low susceptibility to penicillin (16.1%) reflects the typical resistance profile of coagulase-negative staphylococci, which often produce beta-lactamase enzymes. These findings highlight the importance of performing susceptibility testing to tailor antibiotic therapy and avoid the use of ineffective treatments. The preference for fluoroquinolones, often combined with rifampicin, as a treatment regimen reflects current best practise for managing biofilm-related infections [14].

The mortality related to *S. caprae* infections was very low, probably due to the average age of the patients coupled with optimal surgical and antibiotic treatment. Despite the severity of these infections, the study’s low mortality rate, with only one death reported due to prosthetic infective endocarditis, suggests that prompt and aggressive treatment can lead to favourable outcomes. These data are in line with previous studies where only two deaths were reported related to one clinical case of lumbar spondylodiscitis and another clinical case of mastoiditis [20,21].

The present study has several limitations, mainly due to its retrospective design. Therefore, to analyze genomic characteristics of *S. caprae* such as agrD sequence typing was not possible. The unavailability of some information about which antibiotic treatment was prescribed, the lack of uniformity in the duration of some antibiotic treatment prescribed by different specialists and the relatively short study period make it difficult to draw significant conclusions. Prospective studies with larger cohorts are needed to validate these results and provide more definitive conclusions about the epidemiology and management of *S. caprae* infection.

## 5. Conclusions

In conclusion, there are currently few studies describing *S. caprae* infections in Spain, despite a significant increase in the number of cases in recent years. In the data examined here, most patients from rural areas had had livestock exposure, while this was not the case for patients who lived in an urban setting, showing the importance of information about patients’ occupational field in any investigation of this issue. Skin or soft subcutaneous tissue infections and osteoarticular infections were the most prevalent manifestations of *S. caprae* infection in our series, and our data showed a low mortality rate. Ongoing surveillance and research are essential to improve our understanding of this emerging pathogen in order to optimize patient care.

## Figures and Tables

**Figure 1 microorganisms-13-00053-f001:**
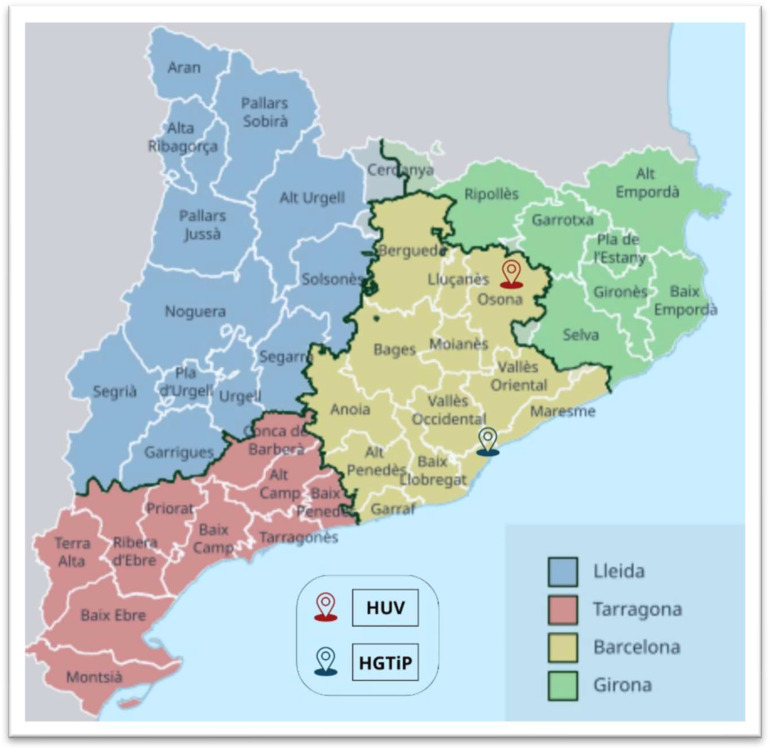
Map of the administrative subdivisions of Catalonia, Spain, showing the locations of the Hospital Universitari de Vic (HUV) and Hospital Germans Trias i Pujol (HGTiP).

**Figure 2 microorganisms-13-00053-f002:**
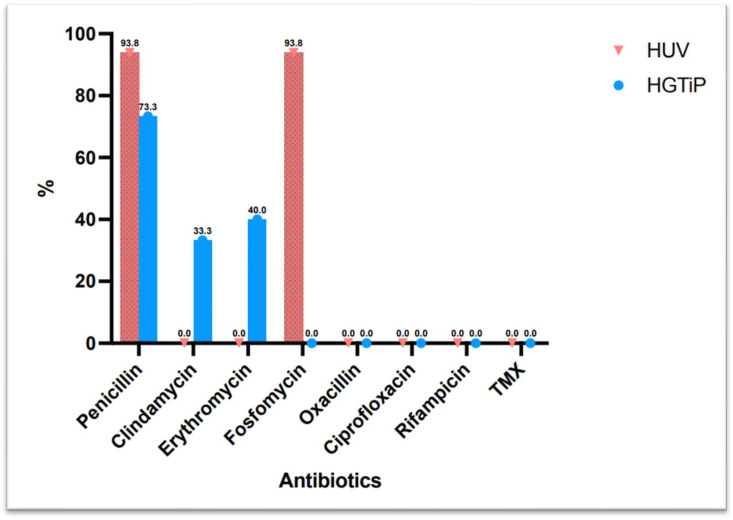
Difference in antibiotic resistance between isolates from two hospitals in a rural and urban area in Catalonia.

**Table 1 microorganisms-13-00053-t001:** Clinical and epidemiological characteristics of patients with *S. caprae* infections at two hospitals in Catalonia.

Characteristics	Total n = 31	HUV n = 16 (51.6%)	HGTiP n = 15 (48.4%)	*p*
Demographic				
Age, mean ± DE	58.87 ± 0.7	55.31 ± 2.4	62.67 ± 8.6	0.981
Male sex	18 (58.1%)	7 (43.8%)	11 (73.3%)	0.095
Period				0.002
2010–2016	8 (25.8%)	8 (50.0%)	0 (0)	
2017–2023	23 (74.2%)	8 (50.0%)	15 (100)	
Charlson Index, median (P25–P75)	3.00 (0.0–5.0)	4.00 (1.0–5.0)	2.50 (0.0–5.0)	0.559
Livestock exposure				0.037
(goats, sheep, pigs, abattoirs, and close relatives)	8 (25.8%)	7 (43.8%)	1 (6.7%)	
Type of infection				0.427
Skin and SSTI	10 (32.3%)	8 (50.0%)	2 (13.3%)	
Osteomyelitis	6 (19.4%)	0 (0.0%)	6 (40.0%)	
Joint prosthetic infections	5 (16.1%)	1 (6.3%)	4 (26.7%)	
Bursitis	3 (9.7%)	1 (6.3%)	2 (13.3%)	
Infective endocarditis	2 (6.5%)	2 (12.5%)	0 (0.0%)	
Urinary tract infection	2 (6.5%)	2 (12.5%)	0 (0.0%)	
Diabetic foot infection	1 (3.2%)	0 (0.0%)	1 (6.7%)	
Tunnelled catheter for haemodialysis infection	1 (3.2%)	1 (6.3%)	0 (0.0%)	
Otitis media	1 (3.2%)	1 (6.3%)	0 (0.0%)	
Oxacillin susceptibility	31 (100.0%)	16 (100.0%)	15 (100.0%)	–
Clindamycin susceptibility	26 (83.9%)	16 (100.0%)	10 (66.7%)	0.018
Penicillin susceptibility	5 (16.1%)	1 (6.3%)	4 (26.7%)	0.172
Outcomes				
In-hospital death	1(3.2%)	1 (3.2%)	0 (0.0%)	1.000
1-year death	0 (0.0%)	0 (0.0%)	0 (0.0%)	–

HUV: Hospital Universitari de Vic; HGTiP: Hospital Germans Trias i Pujol. *p* value for comparison of values across hospitals.

**Table 2 microorganisms-13-00053-t002:** Summary of individual characteristics of the 31 patients diagnosed with *S. caprae* infections at two hospitals in Catalonia.

Case	Hospital and Date	Age (Years)	Sex	Underlying Conditions	Livestock Exposure	Diagnosis	Site of *S. caprae* Isolation	Antibiotic Treatment and Duration	Surgery	Outcome
1	HUV (2021)	37	F	Brugada syndrome	No	Pyomyositis leg	Wound fluid	LZD—28 d	Yes	Cured
2	HUV (2021)	50	F	HTN, TOF, pulmonary prosthesis	Yes	Pulmonary prosthetic valve infective endocarditis	Blood culture	CLX + RIF (42 d) → LEV + RIF	No	Cured
3	HUV (2021)	68	F	HTN, obesity, CRF	Yes	Tunnelled catheter for haemodialysis infection	Wound fluid, catheter culture	Unknown	No (catheter removal)	Cured
4	HUV (2020)	35	M	No	Yes	Gluteal abscess	Purulent discharge	AMO/CV—7 d	Yes	Cured
5	HUV (2019)	77	F	DM, HTN, CRF, breast cancer,	No	Leg ulcer infection	Wound fluid	CIP—14 d	No	Cured
6	HUV (2019)	75	M	DM, HTN, ICM, below-knee amputation of the left lower limb, aortic prosthesis	Yes	Aortic prosthetic valve infective endocarditis	Blood culture	CLX + RIF (42 d) + GEN (14 d)	Yes	Died
7	HUV (2019)	57	M	HTN	Yes	Hip prosthesis infection	Synovial fluid	LEV + RIF—84 d	Yes	Cured
8	HUV (2017)	39	M	No	Yes	Bursitis knee	Bursa	Unknown	Yes	Cured
9	HUV (2016)	64	F	DM, obesity, psoriasis	No	Cellulitis leg	Wound fluid	CD—21 d	No	Cured
10	HUV (2015)	46	F	HIV	Yes	Cellulitis toe	Purulent discharge	TMX—14 d	No	Cured
11	HUV (2013)	83	F	HTN, dementia	No	UTI	Urine culture	Unknown	No	Cured
12	HUV (2012)	85	F	HTN, ICM	No	Leg ulcer infection	Wound fluid	Unknown	No	Cured
13	HUV (2012)	45	F	No	No	Breast abscess	Purulent discharge	AMO/CV—7 d	Yes	Cured
14	HUV (2012)	45	M	Hydrosadenitis	No	Gluteal abscess	Purulent discharge	Unknown	Yes	Cured
15	HUV (2012)	78	M	HTN, cerebral vasculopathy	No	UTI	Urine culture	Unknown	No	Cured
16	HUV (2010)	1	F	No	No	Otitis media	Ear drainage	AMO/CV—14 d	No	Cured
17	HGTiP (2023)	71	F	No	Yes	Osteomyelitis calcaneus	Bone biopsy	AMO/CV—28 d	Yes	Cured
18	HGTiP (2023)	30	M	No	No	Osteomyelitis tibia	Bone biopsy	LEV—28 d	Yes	Cured
19	HGTiP (2023)	68	F	HTN, cerebral vasculopathy	No	Elbow bursitis	Purulent discharge	LEV—28 d	Yes	Cured
20	HGTiP (2023)	80	M	DM, CRF, ICM	No	Osteomyelitis toe	Bone biopsy	LEV—15 d	Yes	Cured
21	HGTiP (2023)	62	M	DM, ICM	No	Cellulitis toe	Skin biopsy	AMO/CV—14 d	Yes	Cured
22	HGTiP (2023)	66	M	HTN, DM, CRF	No	Osteomyelitis toe	Bone biopsy	LEV—28 d	Yes	Cured
23	HGTiP (2023)	80	M	HTN, DM, CRF, ICM, maxillary cancer	No	Osteomyelitis toe	Bone biopsy	LEV—15 d	Yes	Cured
24	HGTiP (2023)	52	M	No	No	Tibial osteosynthesis material infection	Osteosynthesis material culture, bone biopsy	LEV + RIF—56 d	Yes	Cured
25	HGTiP (2023)	68	F	HTN, cerebral vasculopathy	No	Elbow bursitis	Bursa	LEV—28 d	Yes	Cured
26	HGTiP (2023)	30	M	No	No	Tibial osteosynthesis material infection	Osteosynthesis material culture, bone biopsy	LEV—56 d	Yes	Cured
27	HGTiP (2021)	84	M	HTN, DM	No	Astragalus osteosynthesis material infection	Osteosynthesis material culture	Unknown	Yes	Cured
28	HGTiP (2023)	79	M	HTN, DM	No	Knee prosthesis infection	Purulent discharge	LEV—56 d	Yes	Cured
29	HGTiP (2021)	49	M	HTN, DM, ICM, PAD	No	Diabetic foot infection	Purulent discharge	LEV—42 d	Yes	Cured
30	HGTiP (2020)	82	F	DM, Dementia, CRF, PAD	No	Osteomyelitis calcaneus	Purulent discharge	CIP—15 d	No	Cured
31	HGTiP (2018)	39	M	No	No	Finger abscess	Purulent discharge	AMO/CV—14 d	Yes	Cured

List of abbreviations: HUV: Hospital Universitari de Vic; HGTiP: Hospital Germans Trias i Pujol; F: female; M: male; HTN: hypertension; DM: diabetes mellitus; TOF: tetralogy of Fallot; CRF: chronic renal failure; ICM: ischemic cardiomyopathy; UTI: urinary tract infection. PAD: peripheral arterial disease. LZD: linezolid; CLX: cloxacillin; RIF: rifampicin; LEV: levofloxacin; CIP: ciprofloxacin; GEN: gentamicin; AMO/CV: amoxicillin/clavulanic acid; TMX; trimethoprim–sulfamethoxazole; CD: clindamycin; d: days.

## Data Availability

The original contributions presented in this study are included in the article/Appendix A. Further inquiries can be directed to the corresponding authors.

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
