# Peer review of "Clinical and Epidemiological Characteristics of Staphylococcus caprae Infections in Catalonia, Spain"

_microorganisms, 2025, doi:10.3390/microorganisms13010053_

Round 1

Reviewer 1 Report

Comments and Suggestions for Authors

1. There is no mention that this paper exists as a preprint online. 

2. Reference 9 is out-of-context- self-citation. 

3. In the abstract the authors state that most cases involved livestock exposure. 8/31 is not "most" though. 

4. In the Description of the Study section, mentioning the pig industry is irrelevant. 

5. An expanded Table with data of the patients would be useful, including any results of blood cultures in patients where the pathogen was isolated from other sites, as well as exact antibiotic used and duration. 

6. The rationale of the use of different antibiotics should also be discussed. Who chose what and on what basis. 

7. Can the difference between the two periods in case detection be attributed to improved diagnostic techniques or increased awareness? Has for example the rural hospital expanded, had new departments or covered larger areas for any reason? 

8. Given the rarity of S. caprae in the literature, it would be beneficial to include all series, for example Shuttleworth's 1997. I am not talking about case reports, although this would even further enhance the manuscript. 

Author Response

Reviewer 1:
1.There is no mention that this paper exists as a preprint online. 

Answer: In October, we sent the article to another journal and it was registered with this suggested citation (http://dx.doi.org/10.2139/ssrn.4989639) and after the manuscript was rejected. When we sent the manuscript to ‘Microorganisms Journal’ we used Preprints but it was declined for announcement because our manuscript was found to have been announced on another preprinting platform.

2. Reference 9 is out-of-context- self-citation. 

Answer: We have removed the citation accordingly.

3. In the abstract the authors state that most cases involved livestock exposure. 8/31 is not "most" though. 

Answer: The reviewer is right. We have changed “most” for the exact percentage.

4. In the Description of the Study section, mentioning the pig industry is irrelevant. 

Answer: We have removed this information accordingly.

5. An expanded Table with data of the patients would be useful, including any results of blood cultures in patients where the pathogen was isolated from other sites, as well as exact antibiotic used and duration. 

Answer: We appreciate your suggestion, for this reason, we have designed the Table 2 with the summary of the individual characteristics of the 31 patients diagnosed with S. caprae infections (lines 377-385).

6. The rationale of the use of different antibiotics should also be discussed. Who chose what and on what basis.

Answer: We have observed retrospectively that antibiotic treatment in most severe infections such as infective endocarditis or osteoarticular infections was chosen by infectious diseases specialists. However, other infections were treated by specialists such as surgeons according to their clinical judgement and the medical course of the patient. We have added this as limitation of the study (lines 227-228).

7. Can the difference between the two periods in case detection be attributed to improved diagnostic techniques or increased awareness? Has for example the rural hospital expanded, had new departments or covered larger areas for any reason? 

Answer: Thank you for your question. The diagnostic techniques have been the same for the last 13 years in both hospitals. As a possible explanation, it is important to mention that our team are very involved in “One Health” approach and we have the perception that the number of surgeries is probably increasing as well as the awareness of surgeons to send samples for a correct microbiological diagnosis.

8. Given the rarity of S. caprae in the literature, it would be beneficial to include all series, for example Shuttleworth's 1997. I am not talking about case reports, although this would even further enhance the manuscript. 

Answer: Thank you for your comment. In accordance with your suggestion, we have designed a supplementary Table 1 with all series and case reports published in English, Spanish or French, between 1995-2023 to enhance our manuscript.

Reviewer 2 Report

Comments and Suggestions for Authors

Interesting report, retrospectively analyzing S.caprae infections in a large Spanish region. 

Please see my comments below:

Please clarify whether all cases with S. caprae infection were included in this study, or if there is possibility of missed cases.

Page 5: “All of the S. caprae strains isolated were susceptible to oxacillin, fluoroquinolones, rifampicin and trimethoprim-sulfamethoxazole and 83.9% were susceptible to clindamycin, while only 16.1% were susceptible to penicillin”. And “Twelve patients were treated with fluoroquinolones (two of them associated with rifampicin), 5 pa-tients with amoxicillin/clavulanic acid, 2 patients with trimethoprim-sulfamethoxazole, 2 patients with cloxacillin plus rifampicin, and clindamycin and linezolid were used for the remaining 2 pa-tients, respectively. For 8 patients, no information was available regarding the antibiotic treatment prescribed.” Please correct word formatting and line numbering.

Please write an appropriate title for Tables 1&2.

In results, please provide more information on treatment and outcomes: for how long were antibiotics administered? Which antibiotics were given to the patients who underwent surgery? Which antibiotics were administered to the patient who died?

Discussion does not need to be separated into sections.

"The finding that, overall, 25.8% of patients had had direct or indirect exposure to livestock underscores the zoonotic potential of S. caprae. Such findings highlight the im-portance of considering occupational and environmental factors in the epidemiology of S. caprae infections, particularly in rural areas where human-animal interactions are more common.

Interestingly, we found that animal contact was predominantly reported among pa-tients from the HUV, which serves a more rural catchment area, supporting the notion of distinct transmission dynamics between rural and urban settings. In urban areas, alterna-tive transmission routes, such as through contaminated food products or the environ-ment, should be considered. This warrants further investigation to elucidate the mecha-nisms of infection acquisition, particularly in urban populations" I believe that this part warrants further discussion on the epidemiology and other factors potentially affecting transmission and environmental reservoirs of S. caprae. 

“The mortality related to S. caprae infections was very low, probably due to the average age of the patients coupled with optimal surgical and antibiotic treatment. Despite the severity of these infections, the study’s low mortality rate, with only one death reported due to prosthetic infective endocarditis, suggests that prompt and aggressive treatment can lead to favourable outcomes.” Are there similar studies to compare management, outcomes and mortality?

Author Response

Interesting report, retrospectively analyzing S.caprae infections in a large Spanish region. 

Answer:  Thank you for your positive comment.

Please see my comments below:

Please clarify whether all cases with S. caprae infection were included in this study, or if there is possibility of missed cases.

Answer: In collaboration with the Microbiology department of both hospitals, we have access to all our microbiological cultures, so we have added all our recorded cases to our study.

Page 5: “All of the S. caprae strains isolated were susceptible to oxacillin, fluoroquinolones, rifampicin and trimethoprim-sulfamethoxazole and 83.9% were susceptible to clindamycin, while only 16.1% were susceptible to penicillin”. And “Twelve patients were treated with fluoroquinolones (two of them associated with rifampicin), 5 patients with amoxicillin/clavulanic acid, 2 patients with trimethoprim-sulfamethoxazole, 2 patients with cloxacillin plus rifampicin, and clindamycin and linezolid were used for the remaining 2 patients, respectively. For 8 patients, no information was available regarding the antibiotic treatment prescribed.” Please correct word formatting and line numbering.

Answer: Thank your for your advice. We have modified it accordingly (lines 152-159).

Please write an appropriate title for Tables 1&2.

Answer: Thank you for your comment. We have modified the title of both tables accordingly.

In results, please provide more information on treatment and outcomes: for how long were antibiotics administered? Which antibiotics were given to the patients who underwent surgery? Which antibiotics were administered to the patient who died?

Answer: We appreciate your questions, so we have designed a new Table 2 with all these data collected individually. The patient who died was treated with cloxacillin plus rifampicin (both 6 weeks) plus gentamicin (2 weeks).

Discussion does not need to be separated into sections.

Answer: We have removed the heading of sections accordingly.

"The finding that, overall, 25.8% of patients had had direct or indirect exposure to livestock underscores the zoonotic potential of S. caprae. Such findings highlight the im-portance of considering occupational and environmental factors in the epidemiology of S. caprae infections, particularly in rural areas where human-animal interactions are more common.
Interestingly, we found that animal contact was predominantly reported among patients from the HUV, which serves a more rural catchment area, supporting the notion of distinct transmission dynamics between rural and urban settings. In urban areas, alternative transmission routes, such as through contaminated food products or the environment, should be considered. This warrants further investigation to elucidate the mechanisms of infection acquisition, particularly in urban populations" I believe that this part warrants further discussion on the epidemiology and other factors potentially affecting transmission and environmental reservoirs of S. caprae. 

Answer: We have tried to clarify this point taking into account the limited information available on this issue (lines 188-193).

“The mortality related to S. caprae infections was very low, probably due to the average age of the patients coupled with optimal surgical and antibiotic treatment. Despite the severity of these infections, the study’s low mortality rate, with only one death reported due to prosthetic infective endocarditis, suggests that prompt and aggressive treatment can lead to favourable outcomes.” Are there similar studies to compare management, outcomes and mortality?

Answer: We have revised all the literature and mortality directly related to S. caprae infections only was reported in two cases (excluding our case of infective endocarditis). We have added this data in the lines 221-223 with their respective references (lines 350-355) and in the supplementary Table 1.

Reviewer 3 Report

Comments and Suggestions for Authors

This study aimed to characterize Staphylococcus caprae infections identified in two hospitals located in rural and urban areas of Catalonia, Spain. Specifically, the study sought to compile and analyze data from the clinical records of all S. caprae infections diagnosed over a thirteen-year period from January 2010 to December 2023. This study identified 31 cases over a 13-year period, with a notable increase in the latter half (2017-2023), suggesting that S. caprae may become more prevalent in the future. The manuscript highlights the distinct transmission patterns between rural and urban settings. In rural areas, there was a higher incidence of livestock exposure among patients, whereas urban cases had less clear transmission routes. This underscores the importance of considering occupational and environmental factors in S. caprae infections. The most common infections were skin and soft tissue infections, osteomyelitis, and joint prosthetic infections. Notably, two cases of infective endocarditis were reported, which is rare for S. caprae. All the isolated strains were susceptible to oxacillin, fluoroquinolones, rifampicin, and trimethoprim-sulfamethoxazole, but only 16.1% were susceptible to penicillin. Despite the severity of some infections, the study reported a low mortality rate, with only one death due to prosthetic infective endocarditis. This manuscript explicitly mentions several limitations. These limitations are important to consider when interpreting the results and conclusions of this research because they affect the strength and generalizability of the study's conclusions. However, they highlight areas where further research is needed to fully understand the epidemiology, clinical characteristics, and management of S. caprae infection. 

Major comments

Abstract: The abstract can be improved by adding a brief background statement to contextualize the importance of studying S. caprae infections. In addition, given the importance of antibiotic resistance, a brief description of antibiotic susceptibility findings could be valuable. The text could be strengthened by briefly mentioning the implications of these findings for clinical practice and future research.

Introduction: Please add recent epidemiological data on S. caprae infections globally.

Material and methods:

- A significant increase was observed in the latter half of the study period. Couldn't this reflect an improved detection method instead of a true increase in incidence?

- Did the laboratories of both hospitals use the same methodology for bacterial identification and antibiotic susceptibility since 2010? Or were the bacterial isolates stored at -80 °C and then identified by MALDI-TOF for confirmation of the isolates?

Results

- It is difficult to assess the true significance of S. caprae as a pathogen compared with other bacteria. Were the microorganisms isolated in pure culture from all 31 cases of infection? Please clarify this point in the Results section.

- The title of Table 1 is not appropriate. Consider the following suggestion:

Table 1. Clinical and epidemiological characteristics of patients with S. caprae infections at two hospitals in Catalonia

- Add a footnote to the table explaining the acronym of the hospitals and the statistical tests performed.

- Table 2 is a figure and its title should be below the figure. The title could be:

Figure 1: Difference in antibiotic resistance between isolates from two hospitals in rural and urban areas of Catalonia

- The manuscript describes livestock exposure. What about other potential risk factors, such as immunosuppression, underlying health conditions, or specific occupational risks? Do you have these data?

Discussion:

- Could you please explore the potential reasons for the increase in cases over time more thoroughly?

- This study reported 100% susceptibility to oxacillin, which is unusual for staphylococcal species. While not necessarily inconsistent, this finding is noteworthy and could benefit from further discussion or explanation.

Conclusions: Overall, the conclusions seem valid and are well supported by the data presented. The authors were careful not to overstate their findings and acknowledge the need for further research, which adds to the credibility of their conclusions.

Author Response

This study aimed to characterize Staphylococcus caprae infections identified in two hospitals located in rural and urban areas of Catalonia, Spain. Specifically, the study sought to compile and analyze data from the clinical records of all S. caprae infections diagnosed over a thirteen-year period from January 2010 to December 2023. This study identified 31 cases over a 13-year period, with a notable increase in the latter half (2017-2023), suggesting that S. caprae may become more prevalent in the future. The manuscript highlights the distinct transmission patterns between rural and urban settings. In rural areas, there was a higher incidence of livestock exposure among patients, whereas urban cases had less clear transmission routes. This underscores the importance of considering occupational and environmental factors in S. caprae infections. The most common infections were skin and soft tissue infections, osteomyelitis, and joint prosthetic infections. Notably, two cases of infective endocarditis were reported, which is rare for S. caprae. All the isolated strains were susceptible to oxacillin, fluoroquinolones, rifampicin, and trimethoprim-sulfamethoxazole, but only 16.1% were susceptible to penicillin. Despite the severity of some infections, the study reported a low mortality rate, with only one death due to prosthetic infective endocarditis. This manuscript explicitly mentions several limitations. These limitations are important to consider when interpreting the results and conclusions of this research because they affect the strength and generalizability of the study's conclusions. However, they highlight areas where further research is needed to fully understand the epidemiology, clinical characteristics, and management of S. caprae infection. 

Major comments

Abstract: The abstract can be improved by adding a brief background statement to contextualize the importance of studying S. caprae infections. In addition, given the importance of antibiotic resistance, a brief description of antibiotic susceptibility findings could be valuable. The text could be strengthened by briefly mentioning the implications of these findings for clinical practice and future research.

Answer:  We have tried to improve our abstract following your suggestion.

Introduction: Please add recent epidemiological data on S. caprae infections globally.

Answer: Thank you for your suggestion. There is little information about this data, so we have designed a supplementary Table 1, which includes the countries where cases of S. caprae infections have been reported. Therefore, we have added this information to the introduction following your advice (lines 87-90)

Material and methods:
- A significant increase was observed in the latter half of the study period. Couldn't this reflect an improved detection method instead of a true increase in incidence?

Answer: We appreciate your question and we have confirmed with the Microbiology department of both hospitals that the diagnostic techniques have been the same for the last 13 years. Currently, molecular diagnostic techniques are being used at HGTiP that will improve microbiological diagnosis in the future and we will determine whether there is a stabilisation of cases or an increase in incidence. We have added this to the manuscript (lines 117-118).

Did the laboratories of both hospitals use the same methodology for bacterial identification and antibiotic susceptibility since 2010? Or were the bacterial isolates stored at -80 °C and then identified by MALDI-TOF for confirmation of the isolates?

Answer: Thank you for your question. We have corroborated with the Microbiology department of both hospitals that they used the same methodology for bacterial identification and antibiotic susceptibility since 2010.

Results
- It is difficult to assess the true significance of S. caprae as a pathogen compared with other bacteria. Were the microorganisms isolated in pure culture from all 31 cases of infection? Please clarify this point in the Results section.

Answer: We have designed a new table 2 including the site of isolation of S. caprae. In addition, we have confirmed with Microbiology department of both hospitals that they only reported the isolation of this microorganism if the sample was representative and with an adequate clinical background.

The title of Table 1 is not appropriate. Consider the following suggestion:
Table 1. Clinical and epidemiological characteristics of patients with S. caprae infections at two hospitals in Catalonia.

Answer: Thank for your suggestion. We have modified the title following your suggestion.

Add a footnote to the table explaining the acronym of the hospitals and the statistical tests performed.

Answer: Thank for your suggestion. We have added a footnote following your suggestion.

Table 2 is a figure and its title should be below the figure. The title could be:
Figure 1: Difference in antibiotic resistance between isolates from two hospitals in rural and urban areas of Catalonia

Answer: Thank you for your suggestion. The reviewer is right, so we have modified the title and order of this. Furthermore, we have changed the number of the figure because following the order of figures should be the number 2. 

The manuscript describes livestock exposure. What about other potential risk factors, such as immunosuppression, underlying health conditions, or specific occupational risks? Do you have these data?

Answer: We have designed a new table 2 including all the underlying conditions of every patient. Only 1 patient was under corticosteroid or other immunosupresive treatment at the time of infection.

Discussion:
- Could you please explore the potential reasons for the increase in cases over time more thoroughly?

Answer: We have tried to clarify this point and as a possible explanation, it is important to mention that our team are very involved in “One Health” approach and we have the perception that the number of surgeries is probably increasing as well as the awareness of surgeons to send samples for a correct microbiological diagnosis. We have added this to the discusión (lines 170-172).

This study reported 100% susceptibility to oxacillin, which is unusual for staphylococcal species. While not necessarily inconsistent, this finding is noteworthy and could benefit from further discussion or explanation.

Answer: Thank you for your comment. We have revised the literature and most strains of S. caprae showed less resistance to oxacillin compared to other CoNS such as S. epidermidis. Therefore, we have added this information to the manuscript (lines 207-210) with their respective references [14-18] (lines 334-349).

Conclusions: Overall, the conclusions seem valid and are well supported by the data presented. The authors were careful not to overstate their findings and acknowledge the need for further research, which adds to the credibility of their conclusions.

Answer: We appreciate your insight.

Round 2

Reviewer 1 Report

Comments and Suggestions for Authors

satisfied

Reviewer 3 Report

Comments and Suggestions for Authors

The authors have addressed all comments accordingly. The manuscript has improved and is now acceptable for publication.